# 1 Hydrological drought prediction and its influencing

# **features analysis based on a machine learning model**

- 3 Min Li<sup>1,2</sup>, Yuhang Yao<sup>1</sup>, Zilong Feng<sup>3</sup>, Ming Ou<sup>1</sup>
- <sup>1</sup>College of Hydraulic Science and Engineering, Yangzhou University, Yangzhou, China, 225000
- 5 <sup>2</sup>State Key Laboratory of Water Disaster Prevention, China, 210000
- 6 <sup>3</sup>JiLin Province Water Resource and Hydropower Consultative Company of P.R CHINA, Changchun,
- 7 China, 130012

- *Correspondence to*: Min Li (limintju@126.com)
  - Abstract: Predicting future drought conditions are crucial for effective disaster management. In this study, a machine learning framework is proposed to predict hydrological drought in the Huaihe River Basin, China. The Extreme Gradient Boosting (XGBoost) model is applied to predict four drought categories in 28 grid regions for one-month prediction, using 26 features for monthly and 18 for seasonal predictions. The framework also integrates the Shapley Additive Explanation (SHAP) variable importance index to infer drought prediction features. The model achieves 79.9% accuracy in classifying droughts, with the Standard Precipitation Index (SPI) being the most influential feature. The SHAP values of SPI are 0.360, 0.261, 0.169, and 0.247 for spring, summer, autumn, and winter, respectively. Soil moisture content and evapotranspiration are particularly affected in spring and autumn, while large-scale climatic features are more significant in summer and winter. Overall, this study offers valuable decision support for regional drought management and water resource allocation.
- **Keywords**: XGBoost; SHAP; Drought prediction; SRI; Huaihe River Basin

## 1 Introduction

- Drought is a global disaster characterized by its long duration and extensive impacts, resulting in severe implications for the economy, agriculture, and environment (Fu et al., 2018; Shi et al., 2018; Zhou et al., 2020; 2021). Over the past 20 years, the frequency and severity of global drought events have increased (Dai 2011; 2012; 2013; Zhang et al., 2019), affecting water security, economic growth, and food supply in some areas. Therefore, drought prediction is of great significance for managing water resources and reducing losses caused by drought.
- Consequently, according to the different effects of drought, previous studies have divided it into

several different types. Among them, four types of droughts are widely used: meteorology, hydrology, agriculture, and social economy (Wilhite and Glantz, 1985; American Meteorological Society, 2013). In the past few decades, more than one hundred drought indices based on single or multiple hydroclimatic variables have been proposed to represent different drought characteristics. For example, the Palmer Drought Severity Index (PDSI) (Palmer 1965), the Standardized Precipitation Index (SPI) (McKee et al., 1993), and the Standardized Runoff Index (SRI) (Shukla and Wood, 2008). SPI index and SRI index are robust, statistically straightforward to compute, and well-suited to long-term time series data. Therefore, this study chooses the SPI index and SRI index to characterize meteorological drought and hydrological drought.

In recent years, there has been an increasing trend toward utilizing machine learning to predict droughts (Ardabili et al., 2020; Sun and Scanlon, 2019). Compared to conventional regression models, machine learning-based models better capture the non-linear characteristics inherent in drought problems and exhibit more robustness, especially when dealing with high-dimensional datasets (Mishra and Singh, 2010; Kikon and Deka, 2022; Prodhan et al., 2022; Wu et al., 2022). Multiple machine learning models such as artificial neural networks (Orimoloye et al., 2021; Orimoloye et al., 2022), support vector machines (Li et al., 2021), random forests(Park et al., 2019), and extreme gradient boosting (XGBoost) (Choi et al. 2018; Han et al. 2019; Zhang et al., 2023) have been extensively employed in the research field of drought. Machine Learning models can learn the input-output relationships in training data and can effectively leverage big data to improve prediction accuracy (Mardian et al., 2023). By training treebased machine learning models, Bachmair et al. (2016) discovered that tree-based machine learning models outperform baseline models. Jungho and Kim (2023) employed a tree-structured XGBoost model to predict the likelihood of impact occurrence (LIO) of drought on public water supply. Their findings demonstrated that the XGBoost model exhibited high accuracy and low uncertainty. Furthermore, the XGBoost model necessitates only minor hyperparameter tuning, and its performance is relatively insensitive to the selection of hyperparameters (Gao and Ding, 2020; Barnwal et al., 2022).

Previous research indicates that numerous features significantly impact hydrological drought. Zou et al. (2018) demonstrated that climate change is the primary feature affecting hydrological drought on long-term scales. Wang et al. (2021) found that climatic variables such as precipitation and evapotranspiration significantly influence the duration of hydrological drought. Additionally, Gan et al. (2023) revealed that large-scale climatic features and sunspot activity have a substantial impact on

hydrological drought events in the Huaihe River Basin. Despite many studies showing that machine learning models outperform physical models in terms of prediction accuracy, these models lack transparency and interpretability. Most research on machine learning models for drought prediction focuses on model performance, often neglecting the role of different features influencing drought occurrence in model predictions. For example, Xu et al. (2022) established a hybrid model combining autoregressive integrated moving averages (ARIMA) and long short-term memory (LSTM) to predict the standardized precipitation evapotranspiration index at multiple time scales. Yu et al. (2023) combined the Hydrologiska Byrans Vattenbalansavdelning (HBV) model with an LSTM neural network to improve the prediction ability for semi-arid basins. Yalcin et al. (2023) proposed a hybrid model of convolutional neural networks (CNN) and LSTM to enhance the prediction accuracy of the standardized precipitation evapotranspiration index. However, these studies do not consider the influence of different features on the model output.

Recent advancements in Explainable AI (XAI) techniques have provided opportunities for understanding why models make certain predictions (Gunning et al., 2019; Islam et al., 2022). Recently, local interpretability methods have been developed and can be implemented for neural network and random forest model architectures (Ribeiro et al., 2016a). The Local Interpretable Model-Agnostic Explanation (LIME) method has been widely used, but it exhibits a high degree of instability due to considerable variation in its explanations upon repeated use (Ribeiro et al., 2016b). Therefore, the Shapley Additive Explanations (SHAP) approach was proposed as a solution. Grounded in the strong theoretical basis of game theory, it provides more robust mathematical accuracy and consistent extension on top of the LIME framework (Lundberg and Lee, 2017; Molnar, 2022). At present, SHAP has been applied to a variety of prediction scenarios. For example, Dikshit and Pradhan (2021) employed an LSTM model combined with the SHAP algorithm to predict droughts, demonstrating that the inclusion of climate variables as predictors can enhance prediction accuracy. Mardian et al. (2023) utilized an XGBoost model and SHAP to predict droughts in the Canadian prairies, and clarified the importance of spatial and temporal predictors, drought indicators, GRACE groundwater distribution and teleconnection in drought prediction. Similarly, Xue et al. (2024) analyzed the spatial and temporal characteristics and driving factors of agricultural drought during the extreme drought period in northern Italy in 2022 by using the integrated machine learning model explained by SHAP combined with the new integrated agricultural drought index (IADI), quantified the dominant factors, and revealed that meteorological

conditions were the main driving factors. Likewise, Zeng et al. (2025) used the XGBoost model explained by SHAP combined with the new rate of extension (RE) index to analyze the spatial and temporal evolution of meteorological drought characteristics in the Yangtze River Basin of China, quantified the dominant driving factors, and revealed that soil moisture was a primary factor. However, the range of drought-influencing features considered in their research is still not comprehensive enough. For example, soil temperature and water content, surface thermal radiation and other features are also important features affecting drought (Raposo et al., 2023).

In light of the above, the novelty of this study is to employ interpretable machine learning models for hydrological drought prediction and to identify the contribution of different influencing features to the model prediction results. While SPI is a precursor to SRI, this study disentangles the hierarchy of contributing features, including SPI, large-scale climate indices, soil moisture etc. Soil moisture directly affects hydrological drought, and it can analyze the contribution of different features to drought when it is predicted together with drought features such as large-scale climate features. For example, Mardian et al. (2023) employed a method combining the XGBoost model with SHAP (Shapley Additive Explanations) values, utilizing a variety of drought-influencing features such as large-scale climatic features and soil moisture, to predict drought conditions in the context of the Canadian Drought Monitor (CDM) and to understand the underlying driving features. Therefore, the objectives of the study are: i) Utilizing the XGBoost model, combined with 26 features predicted monthly and 18 features predicted seasonally, the hydrological drought in the Huaihe River Basin is predicted, and the performance evaluation is carried out by using precision and recall indicators; ii) Various SHAP plots were employed to gain insights into the model outputs and analyze the influence of different drought variables on the predictive results of the model.

## 2 Study area and data

## 2.1 Study area

In this paper, as shown in Figure 1, the Huaihe River Basin is selected as the research area, and the grid is divided at a resolution of 1°lat×1°lon, with a total of 28 grid regions, which takes into account the computational feasibility and spatial heterogeneity. Although large-cale climatic features have spatial consistency, their effects on regional precipitation can be different through local terrain-atmosphere

feedback (Lu et al., 2006). Gridded analysis identifies sensitive subregions, supporting targeted mitigation. The Huaihe River Basin is located at 111°55'-121°25'E, 30°55'-36°36'N, covering an area of approximately 270,000 square kilometers. It experiences significant spatiotemporal variations in precipitation, with an average annual precipitation of around 883 millimeters. Situated in the transitional climatic zone from south to north, the southern part of the basin falls under a subtropical climate, while the northern part experiences a warm temperate climate. The average annual temperature ranges from 11 to 16°C. The winter and spring seasons in the basin are relatively dry, while the autumn and summer seasons are hot and rainy, resulting in pronounced seasonal fluctuations between droughts and floods. The average annual runoff depth in the basin is 230 millimeters. Due to its unique geographical location, the area is prone to frequent flooding, leading to high water levels and prolonged flood conditions. In addition, the annual average water surface evaporation in the Huaihe River Basin ranges from 900 to 1500 millimeters. As one of the important agricultural production bases in China, the basin is densely populated with substantial water demands. However, the region frequently suffers from drought disasters. Since the beginning of the 21st century, an average of 2.698 million hectares of crops, accounting for 21% of the total cultivated land area in the basin, have been affected annually. The Huaihe River Basin is a significant agricultural area and a high-population-intensive area in eastern China. Seasonal droughts frequently affect food production and water resources. One-month advance prediction is essential for reservoir scheduling, irrigation planning and early warning times for farmers.

Figure 1: Huaihe River Basin and 28 grid area location.

#### 2.2 Data

We obtained monthly average precipitation, wind speed, temperature, evapotranspiration, monthly average runoff, 0-10cm soil moisture, and 100-200cm soil moisture data sets for the Huaihe River Basin from the website <a href="https://disc.gsfc.nasa.gov/datasets/GLDAS\_NOAH10\_M\_2.0/">https://disc.gsfc.nasa.gov/datasets/GLDAS\_NOAH10\_M\_2.0/</a> for the period 1960 to 2014. The monthly average 2 m dewpoint temperature, surface net solar radiation, surface net thermal radiation, surface pressure, and leaf area index data sets were obtained from the ERA5-Land reanalysis dataset (<a href="https://cds.climate.copernicus.eu/">https://cds.climate.copernicus.eu/</a>). According to whether the grid center point falls within the basin, 28 grid regions are defined. If the center point of the grid is not within the basin boundary, the region is not divided into grids. The grid analysis is carried out with these grid points as the center and 1°lat×1°lon as the resolution, covering a total of 28 grid regions. Using the interpolation method based on the Xarray package, the data of Huaihe River Basin are interpolated to 28 grid regions.

Numerous studies have demonstrated the significant influence of large-scale climate indices, including the Atlantic Multidecadal Oscillation (AMO), Arctic Oscillation (AO), North Pacific pattern (NP), Pacific Decadal Oscillation (PDO), and Nino3.4, on drought dynamics(Gan et al., 2023; Phan-Van et al., 2022; Wu and Xu, 2020; Xiao et al., 2019). For example, the positive phase of AMO leads to a decrease in summer precipitation in the Huaihe River Basin by enhancing the western Pacific subtropical

high (Lu et al., 2006); the Pacific Decadal Oscillation (PDO) has the most significant impact on the monthly runoff in the Huaihe River Basin (Sun et al., 2018). These selected climate features (Nino3.4, AMO, TPI, PDO, AO, TNI, and NP) for the Huaihe River basin analysis were acquired from the National Oceanic and Atmospheric Administration (NOAA) climate database (<a href="http://www.esrl.noaa.gov/psd/data/climateindices">http://www.esrl.noaa.gov/psd/data/climateindices</a>), covering the period from 1960 to 2014.

## 3 Methods

## 3.1 Drought indices

In this study, the standardized precipitation index (SPI) (McKee et al., 1993) is used to characterize meteorological drought. SPI is widely used for drought risk assessment and monitoring due to its ease of calculation and ability to work on multiple time scales.

The standardized runoff index (SRI) was first proposed by Shukla and Wood (2008) as an effective and accurate index for describing hydrological drought characteristics. It has been widely used in hydrological drought identification. SRI is also calculated by transforming the cumulative flow distribution of a given time scale into a standard normal distribution using equiprobability transformation, similar to the calculation method of SPI. The SPI/SRI classes are classified as shown in Table 1 (Li et al. 2024). In this study, drought is classified into four classes, namely, Normal (ND), Mild drought (D1), Moderate drought (D2), and Severe drought and Extreme drought (D3), according to Table 1. However, due to the limited number of extreme drought events, it posed an issue in training the model. Therefore, the classes of Severe drought and Extreme drought were merged into one.

Table 1: Drought category classification and corresponding SPI and SRI values.

| SPI/SRI value     | Category      |  |
|-------------------|---------------|--|
| SPI/SRI> 0        | Normal (ND)   |  |
| -1.0≤ SPI/SRI 

Figure 2: Box plots of the accuracy and recall rates of the four drought categories predicted by the 28 regional models ('P' represents the accuracy rate, and 'R' represents the recall rate. The small square represents the average.).

# 4.2 Prediction maps

According to the predicted drought data, 2011 was identified as a year with relatively severe drought conditions. To visually assess the predictive capability of the model, drought predicted, observed, and difference maps were created for each month of 2011 (Figure 3 to Figure 4). Figure 3 shows the comparison between the prediction and observation in the first six months of 2011, and the complete month map is placed in the appendix. In 2011, the model accurately captured drought situations across most regions. In January, the drought situation was severe, and the drought category was mainly in the D2 and D3 categories. However, the prediction map of the model shows that the drought degree in most regions is lighter than the actual drought situation, and the drought category is mainly classified as D1, which relatively underestimates the actual situation of drought. In February, the drought situation was rapidly reduced, and the prediction map of the model was basically consistent with the observation map. In March and April, the drought conditions in the entire basin rapidly escalated and became severe, and

most of the areas in the observation map reached the drought categories of D2 and D3, and only a few areas in the north were classified as D1 drought category. Consequently, this period poses a considerable challenge to the predictive ability of the model, making it an appropriate period to evaluate the predictive performance of the model. In general, the model effectively predicts the occurrence and deterioration of drought and captures the spatial distribution pattern. However, in some parts of the central and western regions, the model still underestimates the drought situation.

Figure 3: Monthly model predictions and observed drought categories for the first six months of 2011.

In May, the severity of the drought situation decreased relative to the previous two months, and the actual observed map and the model-predicted map were largely consistent. According to the observed map, in June, a drought occurrence was observed in the northern region where no drought had been previously recorded. Furthermore, in July, the drought area shifted from the northern to the western region. It was not until August that drought gradually diminished in most areas. Basically, the model captures the change of drought, but for some areas of D3 drought category, the model predicts them as D2 drought category.

In September, drought conditions were found in the eastern and southern regions on the observed map. However, the drought situation in some areas is underestimated on the map predicted by the model.

In October, the model significantly overestimated the severity of the drought situation. According to the observed map, all regions except a small part of the western region experienced the D1 drought category. In contrast, the model-predicted map shows widespread drought across the region, with most of the regions classified in the D2 drought category. In November and December, the drought in the observation map dissipated rapidly, and the drought situation was basically the same as that in the model prediction map.

Figure 4: The difference between the predicted results of the model and the observed data values (Difference = SRI-prediction - SRI-actual) From blue to red indicates that the model predicts the degree of underestimation to overestimation of observations.

In general, the XGBoost model has a great performance in capturing the spatial structure and temporal dynamics of drought events during the 12-month period of 2011. However, the model indicates that while the model can distinguish between drought and non-drought conditions, it lacks clarity in defining the boundaries between different drought categories. In most cases, the model underestimates drought conditions compared to the observed results.

## 4.3 Variable importance analysis

# 4.3.1 Monthly prediction analysis

To study the effects of different features on drought, 26 different drought influencing features were considered, and the corresponding influencing features are analyzed for 28 grid regions, and the contribution analysis is made with SHAP values. Due to the limited space, only the analysis of the 7th grid region is shown in Figure 5. Figure 5 reveals the contribution of each input feature based on the SHAP value of each instance in 28 grid regions. In the vertical direction, the variables in the beeswarm plot are sorted according to their absolute SHAP values, which also reflects the importance of ranking variables. The density of points represents the eigenvalues of each instance in each row. The X-axis shows the SHAP value corresponding to a single instance. The left side of the Y-axis of the bee colony graph represents the negative total contribution of the features in the XGBoost model, while the right side represents the positive total contribution. The negative and positive SHAP values represent the corresponding negative and positive total contribution of the related target variables to the XGBoost model. Therefore, the beeswarm plot reflects the relationship between the variables and the related target variables. The larger the absolute value of SHAP is, the greater the contribution to the model is. The analysis reveals that SPI plays a dominant role, followed by AMO and evapotranspiration.

Figure 5: The SHAP values of 26 different influencing features in each month of the 7th grid region from 2004 to 2014.

To gain a deeper understanding of the features contributing to drought events in the study area, As shown in Figure 6, this study shows the spatial distribution of the first three main drought-influencing features and discusses the changes of drought-influencing features in the basin. The results show that the main influencing feature of hydrological drought in the Huaihe River Basin is meteorological drought. As shown in Table 5, the absolute average SHAP value of the first influencing feature is significantly higher than that of the second and third influencing features. Large-scale climate features (particularly AMO) emerge as the secondary major influence, and about half of the North Central Basin is significantly dependent on these features. For the third influencing feature, a diverse range of large-scale climate variables, such as TPI, PDO, NP, TNI, and AMO, affect almost half of the study area. In summary, the foremost determinant of hydrological drought is meteorological drought. Large-scale climate features (notably AMO) rank second in importance, followed by features like soil moisture content, and so on.

The findings demonstrate that the Standardized Precipitation Index (SPI) serves as the dominant driver of hydrological drought in the Huaihe River Basin, consistent with the conclusions of Gan et al. (2023), who identified meteorological drought as a critical precursor to hydrological extremes in this region. Further support arises from Wang et al. (2021), whose analysis of drought propagation mechanisms in the Huaihe Basin revealed indirect hydrological drought impacts mediated through soil moisture and evapotranspiration—a pattern corroborated by the secondary influence of soil moisture and evapotranspiration in this study. However, compared with the study of Zou et al. (2018) in the Weihe River Basin, the influence of large-scale climate features in this study is more prominent, which may be related to the fact that the Huaihe River Basin is located in the climate transition zone and is more sensitive to the air-sea coupling phenomenon.

Table 5: The first three drought influencing features and the SHAP value of the absolute average influence of 28 grid areas in Huaihe River Basin.

| SHAP<br>value<br>Grid<br>area | The first influencing feature | Average<br>SHAP<br>value | The second influencing feature   | Average<br>SHAP<br>value | The third influencing feature           | Average<br>SHAP<br>value |
|-------------------------------|-------------------------------|--------------------------|----------------------------------|--------------------------|-----------------------------------------|--------------------------|
| 1                             | SPI-1                         | 0.160                    | Evapotranspiration               | 0.040                    | TPI                                     | 0.038                    |
| 2                             | SPI-1                         | 0.190                    | AO                               | 0.018                    | Soil moisture<br>content(100-<br>200cm) | 0.014                    |
| 3                             | SPI-1                         | 0.189                    | TPI                              | 0.030                    | Soil moisture<br>content(100-<br>200cm) | 0.023                    |
| 4                             | SPI-1                         | 0.178                    | NP                               | 0.020                    | PDO                                     | 0.016                    |
| 5                             | SPI-1                         | 0.147                    | Evapotranspiration               | 0.044                    | NP                                      | 0.017                    |
| 6                             | SPI-1                         | 0.180                    | TPI                              | 0.025                    | Evapotranspiration                      | 0.021                    |
| 7                             | SPI-1                         | 0.190                    | AMO                              | 0.037                    | Evapotranspiration                      | 0.023                    |
| 8                             | SPI-1                         | 0.212                    | TPI                              | 0.030                    | TNI                                     | 0.020                    |
| 9                             | SPI-1                         | 0.161                    | AMO                              | 0.034                    | T=2 SPI-6                               | 0.028                    |
| 10                            | SPI-1                         | 0.195                    | AMO                              | 0.037                    | Surface net thermal radiation           | 0.031                    |
| 11                            | SPI-1                         | 0.226                    | AMO                              | 0.037                    | TNI                                     | 0.012                    |
| 12                            | SPI-1                         | 0.221                    | AMO                              | 0.033                    | T=2 SPI-3                               | 0.017                    |
| 13                            | SPI-1                         | 0.228                    | AMO                              | 0.028                    | NP                                      | 0.026                    |
| 14                            | SPI-1                         | 0.204                    | Soil moisture content(100-200cm) | 0.057                    | T=1 SPI-1                               | 0.029                    |

| 15 SPI-1 | CDI 1             | 0.160 | Soil moisture      | 0.033           | 0.033 NP           | 0.032 |
|----------|-------------------|-------|--------------------|-----------------|--------------------|-------|
|          | 51 1-1            |       | content(100-200cm) |                 |                    |       |
| 16       | SPI-1             | 0.157 | Wind speed         | 0.033           | AMO                | 0.030 |
| 17       | SPI-1             | 0.186 | AMO                | 0.064           | Evapotranspiration | 0.025 |
| 10 0     | CDI 1             | 0.225 | AMO                | 0.040           | Soil moisture      | 0.022 |
| 18       | 8 SPI-1 0.235 AMC | AMO   | 0.040              | content(0-10cm) | 0.032              |       |
| 19       | SPI-1             | 0.168 | TPI                | 0.055           | AMO                | 0.035 |
| 20       | SPI-1             | 0.172 | AMO                | 0.038           | T=2 SPI-3          | 0.026 |
| 21       | SPI-1             | 0.165 | AMO                | 0.039           | PDO                | 0.039 |
| 22       | SPI-1             | 0.179 | AMO                | 0.042           | Evapotranspiration | 0.025 |
| 23       | SPI-1             | 0.176 | AMO                | 0.029           | T=1 SPI-9          | 0.022 |
| 24       | SPI-1             | 0.189 | PDO                | 0.053           | AMO                | 0.021 |
| 25       | SPI-1             | 0.149 | AMO                | 0.055           | TPI                | 0.024 |
| 26       | SPI-1             | 0.160 | AMO                | 0.043           | PDO                | 0.030 |
| 27       | SPI-1             | 0.169 | AMO                | 0.047           | T=2 SPI-3          | 0.018 |
| 28       | SPI-1             | 0.287 | NP                 | 0.025           | T=1 SPI-1          | 0.016 |

Figure 6: The first three drought-influencing features of 28 grid areas in the Huaihe River Basin.

# 4.3.2 Seasonal prediction analysis

To accurately reflect the differences in drought-influencing features across different seasons, this study utilized 18 different drought-influencing features to predict the hydrological drought in the Huaihe River Basin. Histograms of the absolute average SHAP values for different influencing features in four seasons in the 7th grid region are presented in Figure 7. The absolute average SHAP values of SPI-3 in spring, summer, autumn, and winter were 0.360, 0.261, 0.169, and 0.247 respectively, which had the greatest impact on hydrological drought in the same season. In addition, the absolute average SHAP values of evapotranspiration, soil moisture content, air temperature, and surface net thermal radiation were close to or exceeded 0.05, which also had a significant impact on hydrological drought in the Huaihe River Basin.

Figure 7: The absolute average SHAP values of 18 different influencing features in the 7th grid region of four seasons ((a) Spring; (b) Summer; (c) Autumn; (d) Winter).

To understand the spatial and temporal distribution characteristics of drought and the potential impact mechanism, Figure 8 displays the spatial distribution of the top three influencing features in each season. The leading influencing features across the four seasons include SPI-3, soil moisture content, and surface net thermal radiation, with SPI-3 being predominant across all seasons and regions. As shown in Figure 9, the absolute average SHAP value of the primary feature exceeded the sum SHAP values of the second and third features. Aside from SPI-3, soil moisture content also exerts a significant influence on hydrological drought in summer and autumn, particularly in the southern and southeastern parts of the river basin. In winter, certain areas in the central part of the river basin are mainly affected by surface net thermal radiation and surface net solar radiation.

From the perspective of the second influencing feature, hydrological drought in most areas of the basin in spring is mainly affected by soil water content and evapotranspiration. In the rest of the region, surface pressure, temperature, radiation, and other features also play an important role. It is worth noting that in the 15th grid region, the surface pressure becomes a key secondary influencing feature, and its absolute average SHAP value reaches 0.175. This value is significantly higher than the second impact feature in other regions, and even close to the primary impact feature in the same grid area. This indicates that it is extremely sensitive to surface pressure in this particular place. During summer, the influence of large-scale climatic features such as the AMO, PDO, and TPI becomes more pronounced compared to spring. Additionally, soil moisture content and surface radiation continue to account for a substantial proportion of the influence on hydrological drought. Regions with absolute average SHAP values surpassing 0.1 in summer constitute approximately one-seventh of the study area, indicating elevated sensitivity to these features during this season. Similar to spring, soil moisture content and evapotranspiration remain predominant influencing features for hydrological drought in half of the grid areas during autumn and winter. The remaining regions are mainly influenced by surface net thermal radiation and surface net solar radiation. Specifically, during winter, the second influencing features for three grid regions (the 12th, 13th, and 21st grid regions) in the central part of the basin are soil moisture content and evapotranspiration, with absolute average SHAP values exceeding 0.1. This indicates a relatively higher influence of these secondary features in these regions compared to others.

Compared with the second impact feature, the large-scale climatic features in the third impact feature have an increased influence on hydrological drought in the four seasons. In spring and autumn, soil moisture content exhibits a more substantial influence on hydrological drought, while in summer, air temperature is considered to be a more important feature. However, in winter, half of the study areas continue to be dominated by soil moisture content and evapotranspiration, whereas most of the remaining study areas are primarily influenced by large-scale climate features features such as TNI, PDO, NP, and AO.

422

Figure 8: The first three drought-influencing features of 28 grid points in Huaihe River Basin in each season.

According to the above results, there were significant differences in the influencing features of drought among the four seasons. This diversity highlights the need for us to pay more attention to the weights and dynamic changes of various influencing features when predicting and understanding the spatial-temporal distribution characteristics of drought. Although the SPI feature continues to dominate, at some grid points, features such as soil moisture content in summer and autumn, as well as thermal radiation in winter, cannot be ignored. This suggests that even for the same influencing feature, its influence can vary greatly in different seasons and regions. Furthermore, in addition to the influence of meteorological drought, the influencing features of spring hydrological drought are mainly biased toward soil moisture content and evapotranspiration, in addition to surface pressure, temperature, radiation, and other related features. The absolute average SHAP value of these influencing features is basically no more than 0.1, which is very different from SPI-3, but its impact on hydrological drought cannot be ignored. In autumn and winter, the above features still dominate, but at the same time, the proportion of large-scale climate features gradually increases, indicating that climate change between different seasons may play an important regulatory role in the composition of drought-influencing features.

Figure 9: The absolute average SHAP values of the first three drought-influencing features in each season.

#### 5 Discussion

This study demonstrates the efficacy of an XGBoost-SHAP framework for hydrological drought prediction in the Huaihe River Basin. The model achieved robust accuracy for the ND and D1 categories, yet underperformed for the more severe categories (D2 and D3), likely due to limited extreme event samples. The prediction of a one-month lead time is helpful for drought monitoring. This enables water managers to adjust reservoir operations and irrigation schedules based on predicted drought conditions. The framework provides a 30-day buffer for proactive measures, such as mobilizing drought relief resources and implementing crop recommendations.

SHAP analysis based on the XGBoost model unequivocally identifies the SPI as the most influential predictor of hydrological drought across the Huaihe River Basin. Such as (Tanriverdi and Batmaz, 2025) for U.S. drought prediction, also identified SPI as one of the most critical features across diverse regions and advanced models. Their SHAP analysis consistently ranked SPI among the top predictors, reinforcing its fundamental role as a primary driver of drought conditions, even within sophisticated deep learning frameworks. Beyond SPI, the key secondary drivers exhibit a distinct spatial and seasonal differences. In terms of space, the hydrological drought in the northern part of the basin shows higher sensitivity to large-scale climate oscillations such as AMO, indicating that large-scale climate features

regulate regional precipitation patterns (Yu et al., 2024). On the contrary, the secondary features affecting the hydrological drought in the southern part of the basin are mainly surface processes, especially soil moisture and evapotranspiration. (Mtupili et al., 2025; Zhu et al., 2025). The difference in the second influencing features of hydrological drought in the southern and northern parts of the basin may be due to the fact that the basin belongs to the temperate-subtropical transition position. For the seasonal scale, in spring, soil moisture and evapotranspiration account for a large proportion of the explanatory power of the model. In summer, the relative weight of large-scale climatic features increases, which is consistent with the enhancement of water vapor transport (Yu et al., 2024). In autumn and winter, radiative fluxes (net solar and thermal radiation) assume greater importance (Jin et al., 2025). Collectively, these findings underscore SPI as the primary driver while revealing the nuanced spatio-temporal controls exerted by secondary features, thereby providing a scientific foundation for developing more targeted drought mitigation and water resource management strategies across the diverse Huaihe River Basin.

When studying the influence of large-scale climate indices on drought, the correlation between climate indices and drought for the same period and a certain lead time is often considered, and the results show that climate indices for the same period and different lead times have a certain influence on drought in the basin, and the degree of influence varies with the changes in the study area. For example, Ren et al. (2017) studied the correlation between SPI and large-scale climate indices with advance periods of 0, 1, 2, and 3 months, and the correlation results show that Nino3.4 has significant correlation in August-October, and PDO has significant correlation in January-May and June-December of the same period. Lv et al. (2022) analyzed the correlation between large-scale climatic features and drought in different lag periods. The results show that large-scale climatic features in the same period also have an impact on drought. Due to the many influencing features considered in this paper, only the effect of climate indices on drought in the basin during the same period was considered when selecting the large-scale climate indices. Subsequent studies can consider selecting the most relevant large-scale climate features in different months or seasons as the influencing features for basin drought prediction to further improve the accuracy of drought prediction. Before inputting the influencing features into the machine learning model for training, methods such as random forest and principal component analysis (PCA) can be used to select the influencing features. Future research can extend the existing one-month-ahead framework to multiple prediction periods to evaluate the impact of different lead times on prediction accuracy. To improve the robustness of the model, a variety of ensemble learning schemes can be compared. Furthermore, the introduction of uncertainty quantification and data enhancement helps to alleviate category imbalances and improve prediction reliability.

#### **6 Conclusions**

Drought is one of the most significant environmental and climate problems in the world, and drought prediction is a crucial means of drought prevention. In this study, the integration of SHAP and XGBoost provides a novel framework that can not only improve the prediction accuracy, but also show the impact of different drought influencing features on drought. The framework can provide two types of support for decision makers: (1) giving priority to high weight features in real-time drought warning; (2) Identifying early risk signals in long-term water resources planning. The main conclusions are as follows:

- 1) The XGBoost model achieved an accuracy of 79.9% for identifying drought categories. The model performs particularly well in predicting ND and D1 drought categories, with a precision rate of 88 % and 74 %, respectively. It also has a recall rate of 91 % and 78 %. However, the prediction performance of the model for the D2 and D3 drought categories is relatively poor, especially for the D3 drought, the recall rate should not exceed 0.5, indicating that the recognition sensitivity of the model for the D3 category is limited. In general, the model has high prediction reliability for ND and D1 categories, but limits in the prediction performance of D2 and D3 categories.
- 2) This study determined that SPI is the most critical factor affecting hydrological drought in the Huaihe River Basin. In 28 grid regions, the absolute average SHAP value of SPI is not less than 0.147, which is much higher than other influencing features. In addition, large-scale climate features, soil moisture content, and evapotranspiration play a significant role in hydrological drought in the basin.
- 3) The SPI remains a major influence in all seasons with absolute average SHAP values of 0.360, 0.261, 0.169, and 0.247 in spring, summer, autumn, and winter respectively. Additional features such as soil moisture content, net heat radiation, and solar radiation also play seasonal roles. Soil moisture content and evapotranspiration are significant features in spring and autumn, while temperature and large-scale climate features are critical in summer and winter.

## 515 S1 Appendices

## 516 Appendix A:

The calculation method of SPI is as follows:

518 
$$f(x) = \frac{x^{\alpha - 1}e^{\frac{x}{\beta}}}{\beta^{\alpha}\Gamma(\alpha)}$$
 (A1)

$$F(x) = \int_0^x f(x) dx \tag{A2}$$

- Assuming that the precipitation series x at a certain time scale follows a stationary gamma
- distribution, where  $\alpha$  and  $\beta$  are the scale and shape parameters ( $\alpha > 0$ ,  $\beta > 0$ ). The cumulative
- probability F(x) of each item is normalized to obtain the corresponding SPI.

# 523 Appendix B:

- Assuming we have K base models denoted as  $f_t(x) \in F$   $t = 1, 2, \dots, K$ , where F the model
- space contains all the base models, the XGBoost model can be represented using the following function:

$$\hat{y} = F(x) = \sum_{t=1}^{k} f_t(x)$$
 (B3)

- Where the parameters of the XGBoost model primarily consist of the structure of each tree and the
- scores in the leaf nodes, that is, the learning of each function  $f_t(x)$
- As each base model is generated in a certain sequential order, the creation of the subsequent tree
- takes into account the predictions made by the preceding tree. Therefore, the objective function of the t
- base model can be expressed as follows:

$$y^{(t)} = \sum_{i=1}^{n} l(y_i, \hat{y}_i^{(t-1)} + f_t(x_i)) + \Omega(f_t)$$
 (B4)

- Here,  $l(y_i, \hat{y}_i^{(t-1)})$  represents the loss function related to  $y_i, \hat{y}_i^{(t-1)}$ ,  $y_i^{(t-1)}$  denotes the
- predictions of the first t-1 decision trees for sample i (i.e., the sum of predictions made by the first
- t-1 trees),  $y_i$  represents the actual value of sample i,  $f_t(x_i)$  represents the prediction of the t
- decision tree for sample i , and  $\Omega(f_t)$  represents the model complexity of the t tree. Therefore,
- the predictions of the first k trees for the sample i are equal to the predictions of the first k-1
- trees plus the prediction of the k tree.

## 539 Appendix C:

The mathematical expression for the classic SHAP value is as follows:

$$\varphi_{i} = \sum_{S \subseteq N} \frac{|S|!(n-|S|-1)!}{n!} \left[ v\left(S \cup \{i\}\right) - v\left(S\right) \right]$$
 (C6)

- Where  $\varphi_i$  represents the contribution of variable i, N represent the set of all variables, n
- denote the number of variables N, S indicate the subset of N that includes variable i, and
- v(N) represent the baseline, which signifies the predicted outcome of each variable in N when their
- values are unknown.
- The model results for each observed value are estimated by summing the SHAP values of each
- variable corresponding to that observed value. Hence, formulating the explanation model as follows:

$$g(z') = \phi_0 + \sum_{i=1}^{M} \phi_i z_i'$$
 (C7)

- Where,  $z' \in \{0,1\}^M$ , the variable quantity is denoted as M, and the value  $\phi_i$  can be obtained from
- equation (C7). SHAP offers a variety of AI model explainers.

## 551 Statements & Declarations

# 552 Software and data availability

- The code and data set for prediction using python language (version 3.9.13) can be found in Mendeley
- Data: doi: 10.17632 / jnr2z36g77.1. The warehouse was created by Yuhang Yao (e-mail: 151746151 @)
- qq.com). The author 's experimental environment is as follows:
- CPU: AMD Ryzen 9 7845HX 3.00 GHz;
- GPU: NVIDIA GeForce RTX4060 8 GB;
- RAM:16G.

#### 559 Funding

- This work was supported by Open Research Fund Program of National Key Laboratory of Water Disaster
- Prevention (No. 2024490711), Yangzhou University Graduate Student Research and Practice Innovation
- Program Funding projects (No. SJCX24\_2250), and Natural Science Foundation of Jiangsu Province
- (No. BK20250906).

**Competing Interests** 565 The authors declare no conflict of interest. 566 **Authors and Affiliations** 567 College of Hydraulic Science and Engineering, Yangzhou University, Yangzhou, China 568 Min Li, Yuhang Yao, Zilong Feng, Ming Ou 569 Corresponding author 570 Correspondence to Min Li, Mail: limintju@126.com 571 **Author Contributions** All authors contributed this paper: Conceptualization, M L; Data curation, M L, YH Y, ZL F, Ming Ou; 572 573 Visualization, YH Y, M L; Validation, M L, YH Y; Methodology, M L, YH Y, ZL F; Formal Analysis, M 574 L, YH Y; Funding acquisition, M L; Writing – original draft, YH Y; Writing – review & editing, M L. 575 Data availability 576 We are grateful to the National Oceanic and Atmospheric Administration ( http://www.esrl.noaa.gov/psd/data/climateindices) for providing large-scale climate index data, and 577 578 grateful to the GLDAS for providing monthly average precipitation, temperature, wind speed, soil water 579 content, evapotranspiration data sets and runoff data sets, and to the European Centre for Medium-Range 580 Weather Forecasts ( https://cds.climate.copernicus.eu/) for providing monthly average 2m dewpoint 581 temperature, surface net solar radiation, surface net thermal radiation, surface pressure, and leaf area 582 index data sets. The data and materials of this study are available. 583 Reference 584 Ardabili, S., Mosavi, A., Dehghani, M., and Várkonyi-Kóczy, A. R.: Deep Learning and Machine 585 Learning in Hydrological Processes Climate Change and Earth Systems a Systematic Review, Cham, 586 52-62, 2020. Bachmair, S., Svensson, C., Hannaford, J., Barker, L. J., and Stahl, K.: A quantitative analysis to 587 588 objectively appraise drought indicators and model drought impacts, Hydrol. EARTH Syst. Sci., 20, 589 2589–2609, https://doi.org/10.5194/hess-20-2589-2016, 2016. 590 Barnwal, A., Cho, H., and Hocking, T.: Survival Regression with Accelerated Failure Time Model in

Graph.

31,

Stat.,

1292-1302.

XGBoost,

J.

Comput.

- https://doi.org/10.1080/10618600.2022.2067548, 2022.
- Chen, T. and Guestrin, C.: XGBoost: A Scalable Tree Boosting System, in: Proceedings of the 22nd ACM
- SIGKDD International Conference on Knowledge Discovery and Data Mining, New York, NY,
- USA, event-place: San Francisco, California, USA, 785–794,
- https://doi.org/10.1145/2939672.2939785, 2016.
- Choi, H.-S., Kim, S., Oh, J. E., Yoon, J. E., Park, J. A., Yun, C.-H., and Yoon, S.: XGBoost-Based
- Instantaneous Drowsiness Detection Framework Using Multitaper Spectral Information of
- Electroencephalography, Washington, DC, USA, https://doi.org/10.1145/3233547.3233567, 2018.
- Dai, A.: Characteristics and trends in various forms of the Palmer Drought Severity Index during 1900-
- 2008, J. Geophys. Res.-ATMOSPHERES, 116, D12115, https://doi.org/10.1029/2010JD015541,
- 2011.
- Dai, A.: Erratum: Drought under global warming: a review, WILEY Interdiscip. Rev.-Clim. CHANGE,
- 3, 617–617, https://doi.org/10.1002/wcc.190, 2012.
- Dai, A.: Increasing drought under global warming in observations and models (vol 3, pg 52, 2013), Nat.
- Clim. CHANGE, 3, 171–171, https://doi.org/10.1038/NCLIMATE1811, 2013.
- Dikshit, A. and Pradhan, B.: Interpretable and explainable AI (XAI) model for spatial drought prediction,
- Sci. TOTAL Environ., 801, 149797, https://doi.org/10.1016/j.scitotenv.2021.149797, 2021.
- Fu, Q., Zhou, Z., Li, T., Liu, D., Hou, R., Cui, S., and Yan, P.: Spatiotemporal characteristics of droughts
- and floods in northeastern China and their impacts on agriculture, Stoch. Environ. Res. RISK
- Assess., 32, 2913–2931, https://doi.org/10.1007/s00477-018-1543-z, 2018.
- Gan, R., Li, D., Chen, C., Yang, F., Zhang, X., and Guo, X.: Spatiotemporal characteristics of extreme
- hydrometeorological events and its potential influencing factors in the Huaihe River Basin, China,
- Stoch. Environ. Res. RISK Assess., 37, 2693–2712, https://doi.org/10.1007/s00477-023-02413-4,
- 2023.
- Gao, L. and Ding, Y.: Disease prediction via Bayesian hyperparameter optimization and ensemble
- learning, BMC Res. NOTES, 13, 205, https://doi.org/10.1186/s13104-020-05050-0, 2020.
- Gunning, D., Stefik, M., Choi, J., Miller, T., Stumpf, S., and Yang, G.-Z.: XAI-Explainable artificial
- intelligence, Sci. Robot., 4, eaay7120, https://doi.org/10.1126/scirobotics.aay7120, 2019.
- Han, Y., Wu, J., Zhai, B., Pan, Y., Huang, G., Wu, L., and Zeng, W.: Coupling a Bat Algorithm with
- KGBoost to Estimate Reference Evapotranspiration in the Arid and Semiarid Regions of China,

- Adv. Meteorol., 2019, 9575782, https://doi.org/10.1155/2019/9575782, 2019.
- Heidke, P.: Berechnung Des Erfolges Und Der Güte Der Windstärkevorhersagen Im
- Sturmwarnungsdienst, Geogr. Ann., 8, 301–349, <a href="https://doi.org/10.1080/20014422.1926.11881138">https://doi.org/10.1080/20014422.1926.11881138</a>,
- 1926.
- Islam, M. R., Ahmed, M. U., Barua, S., and Begum, S.: A Systematic Review of Explainable Artificial
- Intelligence in Terms of Different Application Domains and Tasks, Appl. Sci.-BASEL, 12, 1353,
- <u>https://doi.org/10.3390/app12031353</u>, 2022.
- Jin, H., Zhang, K., Zhang, P., Liu, G., Liu, M., Chen, X., and Willems, P.: Spatiotemporal evolution of
- drought status and its driving factors attribution in China., Sci. Total Environ., 958, 178131–178131,
- https://doi.org/10.1016/j.scitotenv.2024.178131, 2025.
- Jungho, S. and Kim, Y.: Assessing the likelihood of drought impact occurrence with extreme gradient
- boosting: a case study on the public water supply in South Korea, J. HYDROINFORMATICS, 25,
- 191–207, <a href="https://doi.org/10.2166/hydro.2023.064">https://doi.org/10.2166/hydro.2023.064</a>, 2023.
- Kikon, A. and Deka, P. C.: Artificial intelligence application in drought assessment, monitoring and
- forecasting: a review, Stoch. Environ. Res. RISK Assess., 36, 1197-1214,
- https://doi.org/10.1007/s00477-021-02129-3, 2022.
- Li, J., Wang, Z., Wu, X., Xu, C.-Y., Guo, S., Chen, X., and Zhang, Z.: Robust Meteorological Drought
- Prediction Using Antecedent SST Fluctuations and Machine Learning, WATER Resour. Res., 57,
- e2020WR029413, <a href="https://doi.org/10.1029/2020WR029413">https://doi.org/10.1029/2020WR029413</a>, 2021.
- Li, M., Feng, Z., Zhang, M., and Yao, Y.: Influence of large-scale climate indices and regional
- meteorological elements on drought characteristics in the Luanhe River Basin, ATMOSPHERIC
- Res., 300, 107219, https://doi.org/10.1016/j.atmosres.2024.107219, 2024.
- Lu, R., Dong, B., and Ding, H.: Impact of the Atlantic Multidecadal Oscillation on the Asian summer
- monsoon, Geophys. Res. Lett., 33, https://doi.org/10.1029/2006GL027655, 2006.
- Lundberg, S. M. and Lee, S.-I.: A Unified Approach to Interpreting Model Predictions, in: Advances in
- Neural Information Processing Systems, 2017.
- Lv, A., Fan, L., and Zhang, W.: Impact of ENSO Events on Droughts in China, Atmosphere, 13,
- <u>https://doi.org/10.3390/atmos13111764</u>, 2022.
- Mardian, J., Champagne, C., Bonsal, B., and Berg, A.: A Machine Learning Framework for Predicting
- and Understanding the Canadian Drought Monitor, WATER Resour. Res., 59, e2022WR033847,

- https://doi.org/10.1029/2022WR033847, 2023.
- McKee, T. B., Doesken, N. J., and Kleist, J.: THE RELATIONSHIP OF DROUGHT FREQUENCY
- AND DURATION TO TIME SCALES, 1993.
- Mishra, A. K. and Singh, V. P.: A review of drought concepts, J. Hydrol., 391, 204-216,
- <a href="https://doi.org/10.1016/j.jhydrol.2010.07.012">https://doi.org/10.1016/j.jhydrol.2010.07.012</a>, 2010.
- Molnar, C.: Interpretable Machine Learning: A Guide for Making Black Box Models Explainable, 2nd
- ed., 2022.
- Mtupili, M., Wang, R., Gu, L., and Yin, J.: Delayed Response of Soil Moisture and Hydrological
- Droughts to Meteorological Drought Over East Asia, Int. J. Climatol.,
- https://doi.org/10.1002/joc.8883, 2025.
- Orimoloye, I. R., Ololade, O. O., and Belle, J. A.: Satellite-based application in drought disaster
- assessment using terra MOD13Q1 data across free state province, South Africa, J. Environ. Manage.,
- 285, 112112, <a href="https://doi.org/10.1016/j.jenvman.2021.112112">https://doi.org/10.1016/j.jenvman.2021.112112</a>, 2021.
- Orimoloye, I. R., Olusola, A. O., Belle, J. A., Pande, C. B., and Ololade, O. O.: Drought disaster
- monitoring and land use dynamics: identification of drought drivers using regression-based
- digorithms, Nat. HAZARDS, 112, 1085–1106, https://doi.org/10.1007/s11069-022-05219-9, 2022.
- Park, H., Kim, K., and Lee, D. K.: Prediction of Severe Drought Area Based on Random Forest: Using
- Satellite Image and Topography Data, WATER, 11, 705, <a href="https://doi.org/10.3390/w11040705">https://doi.org/10.3390/w11040705</a>, 2019.
- Phan-Van, T., Nguyen-Ngoc-Bich, P., Ngo-Duc, T., Vu-Minh, T., Le, P. V. V., Trinh-Tuan, L., Nguyen-
- Thi, T., Pham-Thanh, H., and Tran-Quang, D.: Drought over Southeast Asia and Its Association with
- Large-Scale Drivers, J. Clim., 35, 4959–4978, https://doi.org/10.1175/JCLI-D-21-0770.1, 2022.
- Prodhan, F. A., Zhang, J., Hasan, S. S., Sharma, T. P. P., and Mohana, H. P.: A review of machine learning
- methods for drought hazard monitoring and forecasting: Current research trends, challenges, and
- future research directions, Environ. Model. Softw., 149, 105327,
- <u>https://doi.org/10.1016/j.envsoft.2022.105327</u>, 2022.
- Raposo, V. de M. B., Costa, V. A. F., and Rodrigues, A. F.: A review of recent developments on drought
- characterization, propagation, and influential factors, Sci. Total Environ., 898, 165550,
- <u>https://doi.org/10.1016/j.scitotenv.2023.165550</u>, 2023.
- Ren, W., Wang, Y., Li, J., Feng, P., and Smith, R. J.: Drought forecasting in Luanhe River basin involving
- climatic indices, Theor. Appl. Climatol., 130, 1133–1148, https://doi.org/10.1007/s00704-016-

- <u>1952-1, 2017.</u>
- Ribeiro, M. T., Singh, S., and Guestrin, C.: Model-Agnostic Interpretability of Machine Learning, 2016a.
- Ribeiro, M. T., Singh, S., and Guestrin, C.: "Why Should I Trust You?": Explaining the Predictions of
- Any Classifier, in: Proceedings of the 22nd ACM SIGKDD International Conference on Knowledge
- Discovery and Data Mining, New York, NY, USA, event-place: San Francisco, California, USA,
- 1135–1144, https://doi.org/10.1145/2939672.2939778, 2016b.
- Shapley, L. S.: 17. A Value for n-Person Games, in: Contributions to the Theory of Games (AM-28),
- Volume II, edited by: Kuhn, H. W. and Tucker, A. W., Princeton University Press, Princeton, 307–
- 318, https://doi.org/doi:10.1515/9781400881970-018, 1953.
- Shi, H., Chen, J., Wang, K., and Niu, J.: A new method and a new index for identifying socioeconomic
- drought events under climate change: A case study of the East River basin in China, Sci. TOTAL
- Environ., 616, 363–375, <a href="https://doi.org/10.1016/j.scitotenv.2017.10.321">https://doi.org/10.1016/j.scitotenv.2017.10.321</a>, 2018.
- Shukla, S. and Wood, A. W.: Use of a standardized runoff index for characterizing hydrologic drought,
- Geophys. Res. Lett., 35, L02405, <a href="https://doi.org/10.1029/2007GL032487">https://doi.org/10.1029/2007GL032487</a>, 2008.
- Sun, A. Y. and Scanlon, B. R.: How can Big Data and machine learning benefit environment and water
- management: a survey of methods, applications, and future directions, Environ. Res. Lett., 14,
- 073001, https://doi.org/10.1088/1748-9326/ab1b7d, 2019.
- Sun, P., Sun, Y., Zhang, Q., and Yao, R.: Hydrological Processes in the Huaihe River Basin, China:
- Seasonal Variations, Causes and Implications, Chin. Geogr. Sci., 28, 636-653,
- <u>https://doi.org/10.1007/s11769-018-0969-z</u>, 2018.
- Tanriverdi, I. and Batmaz, I.: AI-driven US drought prediction using machine learning and deep learning,
- Clim. Dyn., 63, 249, https://doi.org/10.1007/s00382-025-07720-w, 2025.
- Wang, J., Wang, W., Cheng, H., Wang, H., and Zhu, Y.: Propagation from Meteorological to Hydrological
- Drought and Its Influencing Factors in the Huaihe River Basin, WATER, 13, 1985,
- <u>https://doi.org/10.3390/w13141985</u>, 2021.
- Wu, Y. and Xu, Y.: Assessing the Climate Tendency over the Yangtze River Delta, China: Properties,
- Dry/Wet Event Frequencies, and Causes, Water, 12, <a href="https://doi.org/10.3390/w12102734">https://doi.org/10.3390/w12102734</a>, 2020.
- Wu, Z., Yin, H., He, H., and Li, Y.: Dynamic-LSTM hybrid models to improve seasonal drought
- predictions over China, J. Hydrol., 615, 128706, <a href="https://doi.org/10.1016/j.jhydrol.2022.128706">https://doi.org/10.1016/j.jhydrol.2022.128706</a>,
- 2022.

- Xiao, L., Chen, X., Zhang, R., and Zhang, Z.: Spatiotemporal Evolution of Droughts and Their
- Teleconnections with Large-Scale Climate Indices over Guizhou Province in Southwest China,
- Water, 11, https://doi.org/10.3390/w11102104, 2019.
- Xu, D., Zhang, Q., Ding, Y., and Zhang, D.: Application of a hybrid ARIMA-LSTM model based on the
- SPEI for drought forecasting, Environ. Sci. Pollut. Res., 29, 4128–4144,
- https://doi.org/10.1007/s11356-021-15325-z, 2022.
- Xue, C., Ghirardelli, A., Chen, J., and Tarolli, P.: Investigating agricultural drought in Northern Italy
- through explainable Machine Learning: Insights from the 2022 drought, Comput. Electron. Agric.,
- 227, 109572, <a href="https://doi.org/10.1016/j.compag.2024.109572">https://doi.org/10.1016/j.compag.2024.109572</a>, 2024.
- Yalcin, S., Esit, M., and Coban, O.: A new deep learning method for meteorological drought estimation
- based-on standard precipitation evapotranspiration index, Eng. Appl. Artif. Intell., 124, 106550,
- <u>https://doi.org/10.1016/j.engappai.2023.106550</u>, 2023.
- Yu, J., Li, Q., Ding, Y., Wen, Z., Gong, Z., Sun, X., Shen, X., and Dong, L.: AMO modulation of
- interdecadal background of persistent heavy rainfall in summer over the Huaihe River Basin, Clim.
- Dyn., 62, 3621–3640, https://doi.org/10.1007/s00382-023-07088-9, 2024.
- Yu, Q., Jiang, L., Wang, Y., and Liu, J.: Enhancing streamflow simulation using hybridized machine
- learning models in a semi-arid basin of the Chinese loess Plateau, J. Hydrol., 617, 129115,
- https://doi.org/10.1016/j.jhydrol.2023.129115, 2023.
- Zeng, X.-M., Nie, M., Wang, N., Ullah, I., Bai, G., Wang, M., Zhang, Z., and Zhu, J.: Attributing
- spatiotemporal evolution and driving factors of meteorological drought across Yangtze River Basin,
- China, ATMOSPHERIC Res., 323, 108155, https://doi.org/10.1016/j.atmosres.2025.108155, 2025.
- Zhang, B., Abu Salem, F. K., Hayes, M. J., Smith, K. H., Tadesse, T., and Wardlow, B. D.: Explainable
- machine learning for the prediction and assessment of complex drought impacts, Sci. TOTAL
- Environ., 898, 165509, https://doi.org/10.1016/j.scitotenv.2023.165509, 2023.
- Zhang, R., Chen, Z.-Y., Xu, L.-J., and Ou, C.-Q.: Meteorological drought forecasting based on a
- statistical model with machine learning techniques in Shaanxi province, China, Sci. TOTAL
- Environ., 665, 338–346, <a href="https://doi.org/10.1016/j.scitotenv.2019.01.431">https://doi.org/10.1016/j.scitotenv.2019.01.431</a>, 2019.
- Zhou, Z., Shi, H., Fu, Q., Li, T., Gan, T. Y., and Liu, S.: Assessing spatiotemporal characteristics of
- drought and its effects on climate-induced yield of maize in Northeast China, J. Hydrol., 588,
- 125097, https://doi.org/10.1016/j.jhydrol.2020.125097, 2020.

Zhou, Z., Shi, H., Fu, Q., Ding, Y., Li, T., and Liu, S.: Investigating the Propagation From Meteorological 743 to Hydrological Drought by Introducing the Nonlinear Dependence With Directed Information 744 WATER e2021WR030028, Transfer Index, Resour. Res., 57, 745 https://doi.org/10.1029/2021WR030028, 2021. 746 Zhu, S., Huang, W., Wei, Y., Guo, J., and Qin, H.: Impact of driving factors on drought propagation: 747 perspectives on rainfall deficit and excessive evaporation, Clim. Dyn., 63, 204, 748 https://doi.org/10.1007/s00382-025-07618-7, 2025. 749 Zou, L., Xia, J., and She, D.: Analysis of Impacts of Climate Change and Human Activities on Hydrological Drought: a Case Study in the Wei River Basin, China, WATER Resour. Manag., 32, 750

1421–1438, <a href="https://doi.org/10.1007/s11269-017-1877-1">https://doi.org/10.1007/s11269-017-1877-1</a>, 2018.

751