# Peer review of "Hydrological drought prediction and its influencing"

_EGUsphere, 2025_

## Referee Comment (RC1)

egusphere-2025-1891

**"Hydrological drought prediction and its influencing factors analysis based on a machine learning model"**

This manuscript is the latest contribution to a growing amount of literature that seeks to develop machine-learning predictions for droughts. It is an interesting paper that contains an advance analysis on the interpretation of the predictions. Machine learning is, compared to other hazards, under-researched for droughts, and therefore the results of this paper are very relevant to the wider scientific community. However, there are some significant shortcomings that need to be addressed before the manuscript can be considered for publication in NHESS. These main concerns are listed under "general comments".

**General comments:**

1) My main concern comprises the lack of focus on the lead time of the predictions. While the abstract starts with "Predicting future drought conditions", it is unclear how this "future" is represented in the paper. This is a recurring pattern throughout the whole paper, and is missing in all sections, including the methods and results (Figure 2, 3, 4 and 5). It is crucial for the scientific quality and relevance of this paper, that it becomes very clear on which lead time (i.e. how far in future) the model is predicting, and for each Figure to be clear for which lead time the results apply. If only the current timestep is used (i.e. no lead times are implemented), this should be clearly mentioned and reflected upon.

2) While the introduction is generally well-written, the article would benefit from a stronger foundation of the research gap. A lot of attention is now focussed on the explainability of the XGBoost algorithm. I do really not agree with the narrative that XGBoost is very explainable, actually it is a very complex algorithm with many hyperparameters, multiple trees and difficult algorithms (.e.g. tree boosting). On line 76, you state that "At present, there are few studies on interpretable machine learning using the SHAP algorithm." This is not true: SHAP is an extremely popular and frequently used ML algorithm, also in the field of (drought) forecasting. Therefore, instead of this narrative, it would be better to give more examples of previous studies deploying SHAP on ML (drought) predictions (including in the discussion, see point 7, below). Moreover, the research gap should be elaborated based on 1) why ML is used, instead of simpler prediction methods (e.g. linear regression), with examples from previous studies showing that ML has a better performance, 2) more regional context: why is an (earlier) prediction of drought needed in the specific case study region, and 3) what is it exactly that we do not yet know about the drivers of drought, which need to be discovered with the SHAP algorithm.

3) The methods (3.1) should start with a crystal clear overview of the modelling setup (training period, test period, validation period, instead of only in the results, line 250), and all data used. It is now insufficiently clear which variables/data are input to the model (features), and which are output (the target variables), and on which lead times. I propose to make one table including all those components. I think this new Section 3.1 and its table should include all information now listed in Section 3.3, Section 2.2 and Table 1, 2 and 3. Furthermore, it is also unclear how the lead times are implemented, and this aspect should be better represented. Therefore, suggestion to add an extra separate column for "Lead time" (including T=0) and add that the unit is "months"

4) The section about SHAP (Section 3.5) should focus less/not on the formulas, but should better explain the basic principle of SHAP. It should include that SHAP values reflect the 1) local feature importance, instead of global feature importance and 2) that the SHAP value is calculated with reference to a SHAP baseline prediction, and it should be clearly stated which baseline prediction is used.

5) A confusion matrix, showing the number of true/false positives and true/false negatives should definitely be included to shed light on the interpretation of the recall/precision. In case of class imbalance (relatively many true positives), even a useless model could generate high skill scores. I suggest to add another skill score to make your conclusions more robust, such as the Heidke Skil Score.

6) The results can be better visualized with the observations (Figure 3) and predictions (Figure 4) next to each other, in one figure. For this figure, I would suggest to select some months, and move the figure(s) with all months to an appendix).

7) The discussion section should be improved, with a stronger reflection on how the results can lead to better "drought mitigation and water resource management" in practice (including the lead times considered and its implications for proactive/anticipatory drought management, and the consequences/reasons for the relatively poor performance for more extreme drought classes). Moreover, it should provide much more insights into how the results found (e.g. strong importance of SPI) compare to the current state-of-the-art literature.

**Specific comments:**

a. Line 8: suggest to remove "interpretable": XGBoost is not specifically interpretable compared to other decision-tree type algorithms. Whether it is

interpretable, depends on if a thorough analysis has been executed to better understand the predictions

b. Line 9: change "factors" to "features" (also in the rest of the manuscript), and add the 4 drought categories. At a similar note, the "drought impact factors" should be renamed to the "target variable" of the model. The model does not predict real impacts, which is addressed in the general comments section.

c. Line 11: 79.9% accuracy in classifying droughts. On which lead time (i.e. how far in future) are those predictions? This is critical information.

d. Line 132: what do you mean with "Using the interpolation method in array"? Xarray package?

e. Suggestion to put all formulas (except the recall/precision) into the appendix/supplementary material.

f. Line 145: index to indices (plural)

g. Table 1: > 2.0 instead of < 2.0, and abbreviations should be included (as used in the results, e.g. Figure 2).

h. Line 338: More context is needed for the SHAP plot. How can the direction of the relationship of the variables be explained? I see that higher SPI-1 values lead to higher model predictions. Does this mean a higher SPI value leads to stronger drought conditions? Again, it is not entirely clear what the target variable is here.

i. Figure 7: It seems like you can delete this figure, as it is very similar to Figure 6.

j. I really like Table 5 and Figure 8. However, the colors in Figure 8 are not easy to distinguish (e.g. the AMO looks like wind speed..)

k. Figure 11: axis should be labelled.

**Technical corrections:**

a. References should all be double-checked and aligned with the bibliography (e.g. line 27, American Meteorological Society, 2013, is incorrect and not listed).

b. Suggestion to abbreviate machine learning to "ML"

c. Review the article carefully on typos. I found the following:
   a. Line 104: typo in "large-scale
   b. Line 273, "however"

---

## Author Response (AR1)

**Reply on RC1**

This manuscript is the latest contribution to a growing amount of literature that seeks to develop machine-learning predictions for droughts. It is an interesting paper that contains an advance analysis on the interpretation of the predictions. Machine learning is, compared to other hazards, under-researched for droughts, and therefore the results of this paper are very relevant to the wider scientific community. However, there are some significant shortcomings that need to be addressed before the manuscript can be considered for publication in NHESS. These main concerns are listed under "general comments".

**General comments:**

1. My main concern comprises the lack of focus on the lead time of the predictions.

While the abstract starts with "Predicting future drought conditions", it is unclear

how this "future" is represented in the paper. This is a recurring pattern throughout the whole paper, and is missing in all sections, including the methods and results (Figure 2, 3, 4 and 5). It is crucial for the scientific quality and relevance of this paper, that it becomes very clear on which lead time (i.e. how far in future) the model is predicting, and for each Figure to be clear for which lead time the results apply. If only the current timestep is used (i.e. no lead times are implemented), this should be clearly mentioned and reflected upon.

**Respond:** Thank you for your valuable comments. The lead time is 1 month. We illustrate this in Table 2 in Method 3.3.

2. While the introduction is generally well-written, the article would benefit from a stronger foundation of the research gap. A lot of attention is now focused on the explainability of the XGBoost algorithm. I do really not agree with the narrative that XGBoost is very explainable, actually it is a very complex algorithm with many hyperparameters, multiple trees and difficult algorithms (.e.g. tree boosting). On line 76, you state that "At present, there are few studies on interpretable machine learning using the SHAP algorithm." This is not true: SHAP is an extremely popular and frequently used ML algorithm, also in the field of (drought) forecasting. Therefore, instead of this narrative, it would be better to give more examples of previous studies deploying SHAP on ML (drought) predictions (including in the discussion, see point 7, below). Moreover, the research gap should be elaborated based on 1) why ML is used, instead of simpler prediction methods (e.g. linear regression), with examples from previous studies showing that ML has a better performance, 2) more regional context: why is an (earlier) prediction of drought needed in the specific case study region, and 3) what is it exactly that we do not yet know about the drivers of drought, which need to be discovered with the SHAP algorithm.

**Respond:** Thank you for your valuable comments.

- 1) In the abstract section, "The interpretable Extreme Gradient Boosting (XGBoost) model is applied to predict four drought categories in 28 grid regions" been revised as: "The Extreme Gradient Boosting (XGBoost) model is applied to predict four drought categories in 28 grid regions".
- 2) The Shap-related drought prediction research has been supplemented in the introduction: Similarly, Xue et al. (2024) analyzed the spatial and temporal characteristics and driving factors of agricultural drought during the extreme drought period in northern Italy in 2022 by using the integrated machine

learning model explained by SHAP combined with the new integrated agricultural drought index (IADI), quantified the dominant factors, and revealed that meteorological conditions were the main driving factors. Likewise, Zeng et al. (2025) used the XGBoost model explained by SHAP combined with the new rate of extension (RE) index to analyze the spatial and temporal evolution of meteorological drought characteristics in the Yangtze River Basin of China, quantified the dominant driving factors, and revealed that soil moisture was a primary factor.

- 3) It is mentioned in the introduction: Compared to conventional regression models, machine learning-based models better capture the non-linear characteristics inherent in drought problems and exhibit more robustness, especially when dealing with high-dimensional datasets.
- 4) Add to the study area: The Huaihe River Basin is a significant agricultural area and a high-population-intensive area in eastern China. Seasonal droughts frequently affect food production and water resources. One-month advance prediction is essential for reservoir scheduling, irrigation planning and early warning times for farmers.
- 5) It is mentioned in the introduction: While SPI is a precursor to SRI, this study disentangles the hierarchy of contributing features, including SPI, large-scale climate indices, soil moisture etc. Soil moisture directly affects hydrological drought, and it can analyze the contribution of different features to drought when it is predicted together with drought features such as large-scale climate features. For example, Mardian et al. (2023) employed a method combining the XGBoost model with SHAP (Shapley Additive Explanations) values, utilizing a variety of drought-influencing features such as large-scale climatic features and soil moisture, to predict drought conditions in the context of the Canadian Drought Monitor (CDM) and to understand the underlying driving features.
- 3. The methods (3.1) should start with a crystal clear overview of the modelling setup (training period, test period, validation period, instead of only in the results, line 250), and all data used. It is now insufficiently clear which variables/data are input to the model (features), and which are output (the target variables), and on which lead times. I propose to make one table including all those components. I think this new Section 3.1 and its table should include all information now listed in Section 3.3, Section 2.2 and Tables 1, 2 and 3. Furthermore, it is also unclear how the lead times are implemented, and this aspect should be better represented. Therefore, suggestion to add an extra separate column for "Lead time" (including T=0) and add that the unit is "months".

**Respond:** Thank you for your valuable comments. It has been supplemented in Method 3.3. The specific content is as follows:

**3.3 Modeling Settings**

The study period for this research spans from 1960 to 2014, with the model training period from 1960 to 2003 and the prediction period from 2004 to 2014. The input and output data types for 28 grid areas are the same. We use a sliding window of 12 and 3 months. The prediction lead time is 1 month. The relevant settings for models are shown in Table 2.

Take the 7th grid area as an example. When using monthly data, the input was 26 different droughtinfluencing features, and the output was SRI-1. The number of input samples during model training was 13767, and the number of output samples was 526. There are 3432 input samples and 132 output samples during the model prediction period. When using seasonal data, the input is 18 features without drought, and the output is SRI-3 in different seasons. The number of input samples during model training is 792, and the number of output samples is 44. The number of input samples in the model prediction period is 198, and the number of output samples is 11. The model uses Bayesian hyperparameter optimization to find optimal parameters, such as learning rate, tree depth, and number of iterations.

Table 2. Model setup and data overview

| Phase                 | Data Period | Input Window          | Lead time | Output |
|-----------------------|-------------|-----------------------|-----------|--------|
| Training phase        | 1960-2003   | M-12 to M-1 (12month) | 1 month   | SRI-1  |
| (monthly time scale)  |             |                       |           |        |
| Validation phase      | 2004-2014   | M-12 to M-1 (12month) | 1 month   | SRI-1  |
| (monthly time scale)  |             |                       |           |        |
| Training phase        | 1960-2003   | M-3 to M-1 (3month)   | 1 month   | SRI-3  |
| (seasonal time scale) |             |                       |           |        |
| Validation phase      | 2004-2014   | M-3 to M-1 (3month)   | 1 month   | SRI-3  |
| (seasonal time scale) |             |                       |           |        |

Table 3: The monthly scale and seasonal scale of the model predict the input target variables. (T is the lead time, SPI-1, SPI-3, SPI-6, and SPI-9 are SPI values at different monthly scales.).

| Drought influencing features (monthly) | SPI-1, T=1 SPI-1, T=1 SPI-3, T=1 SPI-6, T=1 SPI-9, T=2 SPI-1, T=2         |  |  |  |
|----------------------------------------|---------------------------------------------------------------------------|--|--|--|
|                                        | SPI-3, T=2 SPI-6, T=2 SPI-9, d2m temperature, surface pressure,           |  |  |  |
|                                        | evapotranspiration, Air temperature, wind speed, surface net solar        |  |  |  |
|                                        | radiation, surface net thermal radiation, 0-10cm soil moisture, 100-200cm |  |  |  |
|                                        | soil moisture, Nino3.4, AMO, PDO, AO, TNI, NP, TPI, leaf area index       |  |  |  |
| Drought influencing                    | SPI-3 (different seasons), d2m temperature, surface pressure,             |  |  |  |
| feature (seasonal)                     | evapotranspiration, Air temperature, wind speed, surface net solar        |  |  |  |

radiation, surface net thermal radiation, 0-10cm soil moisture, 100-200cm

soil moisture, Nino3.4, AMO, PDO, AO, TNI, NP, TPI, leaf area index

4. The section about SHAP (Section 3.5) should focus less/not on the formulas, but should better explain the basic principle of SHAP. It should include that SHAP values reflect the 1) local feature importance, instead of global feature importance and 2) that the SHAP value is calculated with reference to a SHAP baseline prediction, and it should be clearly stated which baseline prediction is used.

**Respond:** Thank you for your valuable comments.

- In Method 3.5, add: Importantly, SHAP values reflect local feature importance, meaning that they
  quantify the contribution of each variable to a specific prediction instance, rather than summarizing
  its overall effect across the entire dataset.
- 2) It has been modified in Method 3.5: In our study, the SHAP baseline is the difference between the model prediction and the average prediction of the data set. For each sample and each feature, the SHAP value is the difference between the predicted value of the model containing the feature and the predicted value after removing the feature and the baseline. We use these SHAP values to quantitatively analyze the positive or negative effects of each predictor on hydrological drought prediction.
- 5. A confusion matrix, showing the number of true/false positives and true/false negatives should definitely be included to shed light on the interpretation of the recall/precision. In case of class imbalance (relatively many true positives), even a useless model could generate high skill scores. I suggest to add another skill score to make your conclusions more robust, such as the Heidke Skil Score.

**Respond:** Thank you for your valuable comments. The Heidke Skill Score has been supplemented in Method 3.4.

In result 4.1, add: 'For ND, the HSS is 0.77, showing a significant discriminant advantage over the no-skill baseline that always predicts the most common category.' 'The HSS metric complements precision and recall by evaluating the model's performance relative to the no-skill baseline. Values closer to 1 indicate superior performance. The declining HSS from ND to D2 underscores the model's reduced discriminatory power for less extreme drought categories, aligning with the observed precision-recall trade-offs.'

6. The results can be better visualized with the observations (Figure 3) and predictions (Figure 4) next to each other, in one figure. For this figure, I would suggest to select some months, and move the figure(s) with all months to an appendix).

**Respond:** Thank you for your valuable comments. The first six months of Figure 3 and Figure 4 have been combined and compared. The complete month map has been moved to the appendix.

7. The discussion section should be improved, with a stronger reflection on how the results can lead to better "drought mitigation and water resource management" in practice (including the lead times

considered and its implications for proactive/anticipatory drought management, and the consequences/reasons for the relatively poor performance for more extreme drought classes). Moreover, it should provide much more insights into how the results found (e.g. strong importance of SPI) compare to the current state-of-the-art literature.

**Respond:** Thank you for your valuable comments. It has been added to the discussion section: This study demonstrates the efficacy of an XGBoost-SHAP framework for hydrological drought prediction in the Huaihe River Basin. The model achieved robust accuracy for the ND and D1 categories, yet underperformed for the more severe categories (D2 and D3), likely due to limited extreme event samples. The prediction of a one-month lead time is helpful for drought monitoring. This enables water managers to adjust reservoir operations and irrigation schedules based on predicted drought conditions. The framework provides a 30-day buffer for proactive measures, such as mobilizing drought relief resources and implementing crop recommendations.

In the second paragraph of the discussion section, add: "Such as Tanriverdi and Batmaz (2025) for U.S. drought prediction, also identified SPI as one of the most critical features across diverse regions and advanced models (including LightGBM, LSTM, and Transformer architectures). Their SHAP analysis consistently ranked SPI among the top predictors, reinforcing its fundamental role as a primary driver of drought conditions, even within sophisticated deep learning frameworks."

"Future research can extend the existing one-month-ahead framework to multiple prediction periods to evaluate the impact of different lead times on prediction accuracy. To improve the robustness of the model, a variety of ensemble learning schemes can be compared. Furthermore, the introduction of uncertainty quantification and data enhancement helps to alleviate category imbalances and improve prediction reliability."

**Specific comments:**

a. Line 8: suggest to remove "interpretable": XGBoost is not specifically interpretable compared to other decision-tree type algorithms. Whether it is interpretable, depends on if a thorough analysis has been executed to better understand the predictions

Respond: Thank you for your valuable comments. We have deleted 'interpretable '.

b. Line 9: change "factors" to "features" (also in the rest of the manuscript), and add the 4 drought categories. At a similar note, the "drought impact factors" should be renamed to the "target variable" of the model. The model does not predict real impacts, which is addressed in the general comments section.

**Respond:** Thank you for your valuable comments. We have changed 'factors 'to 'features '; change 'drought impact factors 'to 'target variables '.

c. Line 11: 79.9% accuracy in classifying droughts. On which lead time (i.e. how far in future) are those predictions? This is critical information.

**Respond:** Thank you for your valuable comments. The relevant instructions have been supplemented as follows in the abstract section: The Extreme Gradient Boosting (XGBoost) model is applied to predict four drought categories in 28 grid regions for one-month prediction, using 26 features for monthly and 18 for seasonal predictions.

- d. Line 132: what do you mean with "Using the interpolation method in array"? Xarray package?
  Respond: Thank you for your valuable comments. It has been modified in Section 2.2: Using the interpolation method based on the Xarray package.
- e. Suggestion to put all formulas (except the recall/precision) into the appendix/supplementary material.

**Respond:** Thank you for your valuable comments. We have made the necessary revisions.

f. Line 145: index to indices (plural)

**Respond:** Thank you for your valuable comments. We have made the necessary revisions.

g. Table 1: > 2.0 instead of < 2.0, and abbreviations should be included (as used in the results, e.g. Figure 2).

**Respond:** Thank you for your valuable comments. According to your suggestion, we have made changes as follows:

Table 1: Drought category classification and corresponding SPI and SRI values.

| SPI/SRI value                      | Category      |  |
|------------------------------------|---------------|--|
| SPI/SRI> 0                         | Normal (ND)   |  |
| -1.0≤ SPI/SRI

Figure 6: The first three drought-influencing features of 28 grid areas in the Huaihe River Basin.

k. Figure 11: axis should be labelled.

**Respond:** Thank you for your valuable comments. We have made the necessary revisions.

Figure 9: The absolute average SHAP values of the first three drought-influencing features in each season.

**Reply on RC2**

This paper proposes an interpretable machine learning framework to predict hydrological droughts in the Huaihe River Basin of China, emphasizing the impact of meteorological precursors, large-scale climate indices, and land surface processes across spatial grids and seasons. By integrating 26 influencing factors and quantifying their contributions through SHAP values, this study advances drought prediction beyond conventional statistical models and provides feasible insights for regional water resources management. This work addresses the timely need for interpretable AI in climate risk assessment, and the structure is clear. However, the following points need to be paid attention to strengthen its scientific rigor.

The method partially refers to 'Table 1' (SPI / SRI classification standard), but the subsequent 'Table 2' (monthly scale factor) is mislabeled as 'Table 1'. Please check the full text table number consistency.

**Respond:** Thank you for your valuable comments. We have made the necessary revisions.

2. Figure 5 does not explain the specific meaning of 'positive / negative difference '.

**Respond:** Thank you for your valuable comments. It has been supplemented in the name of the Figure 5.

3. It is only mentioned that XGBoost is superior to the traditional regression model, but it is not compared with the mainstream time series model (such as LSTM). Please briefly explain the reason for choosing XGBoost instead of time series model.

**Respond:** Thank you for your valuable comments. While LSTM and similar time-series models are effective for sequential data, our choice of XGBoost was driven by: 1. Significantly lower computational requirements for operational prediction. 2. Compatibility with SHAP interpretation.

4. Table 1 currently lists the drought level. To avoid ambiguity, please include a precise SPI / SRI value range.

**Respond:** Thank you for your valuable comments. It has been modified in Table 1 of Method 3.1.

5. Section 3.2 describes the goals of XGBoost, but omits the specific hyperparameters and how they are selected. Please supplement the hyperparameter tuning method.

**Respond:** Thank you for your valuable comments. It has been supplemented in Method 3.3: The model uses Bayesian hyperparameter optimization to find optimal parameters, such as learning rate, tree depth, and number of iterations.

6. Figure 1, why the authors delineate the Basin into 28 grids? Some regions are not numbered in this figure.

**Respond:** Thank you for your valuable comments. We divided the entire basin into a 1  $^{\circ} \times 1$   $^{\circ}$  grid. Whether each region is numbered depends on whether the center is within the basin

7. The study period is 1960-2014, can you collect the recent data?

**Respond:** Thank you for your valuable comments. There is no closer data for the time being, and closer data can be supplemented in future studies.

8. In the figures, only the 7th grid region is displayed. Can you provide the results of other regions for comparison?

**Respond:** Thank you for your valuable comments. Due to the limitation of text size, we only put the relevant pictures of 7th. Although we did not show other regions in the form of a graph, table 6 shows the first three drought influencing features and the SHAP value of the absolute average influence of 28 grid areas in Huaihe River Basin. The first three drought influencing features are crucial to drought prediction.

9. Discussion part is very simple. Please discuss the result and limitations, future works deeply.

Respond: Thank you for your valuable comments. It has been supplemented in the discussion: This study demonstrates the efficacy of an XGBoost-SHAP framework for hydrological drought prediction in the Huaihe River Basin. The model achieved robust accuracy for the ND and D1 categories, yet underperformed for the more severe categories (D2 and D3), likely due to limited extreme event samples. The prediction of a one-month lead time is helpful for drought monitoring. This enables water managers to adjust reservoir operations and irrigation schedules based on predicted drought conditions. The framework provides a 30-day buffer for proactive measures, such as mobilizing drought relief resources and implementing crop recommendations.

In the second paragraph of the discussion, add: 'Such as Tanriverdi and Batmaz (2025) for U.S. drought prediction, also identified SPI as one of the most critical features across diverse regions and advanced models (including LightGBM, LSTM, and Transformer architectures). Their SHAP analysis consistently ranked SPI among the top predictors, reinforcing its fundamental role as a primary driver of drought conditions, even within sophisticated deep learning frameworks.'

'Future research can extend the existing one-month-ahead framework to multiple prediction periods to evaluate the impact of different lead times on prediction accuracy. A variety of ensemble learning

schemes can be compared to explore ways to improve the robustness of the model. At the same time, the introduction of uncertainty quantification and data enhancement helps to alleviate category imbalances and improve prediction reliability. The application of these methods provides strong support for more accurate drought trend prediction and management strategies.'